

# Detailed investigation of multi-scale fracture networks in glacially abraded crystalline bedrock at Åland Islands, Finland

Nikolas Ovaskainen[1,2], Pietari Skyttä[2], Nicklas Nordbäck[1,2], and Jon Engström[1]

[1]Geological Survey of Finland, P.0. Box 96, Espoo, FI-02151, Finland
[2]University of Turku, Department of Geography and Geology, Finland

**Correspondence:** Nikolas Ovaskainen (nikolas.ovaskainen@gtk.fi)

**Abstract.** Using multiple scales of observation in studying the fractures of the bedrock increases the reliability and representativeness of the respective studies. This is because the discontinuities, i.e., the fractures, in the bedrock lack any characteristic length and instead occur within a large range of scales of approximately 10 orders of magnitude. Consequently, fracture models need to be constructed based on representative multi-scale datasets to enable valid interpolation and extrapolation of common

scaling laws to all fracture sizes.

In this paper, we combine a detailed bedrock fracture study from an extensive bedrock outcrop area with lineament interpretation using Light Detection And Ranging (LiDAR) and geophysical data. Our study offers lineament data in an intermediary length range missing from Discrete Fracture Network -modelling conducted at Olkiluoto, a nuclear spent fuel facility in Finland. In addition, this study also provides a robust multi-scale fracture and lineament dataset which has been thoroughly

analysed for the purposes of understanding the uncertainties and differences in the different datasets. Our analysis further covers the topological, scale-independent, fracture network characteristics.

Results of our study include the discovery of three distinct azimuth sets, N-S, NE-SW and WNW-ESE, both single scale and multi-scale power-law models for fracture and lineaments and further insight into a trend of decreasing apparent connectivity of fracture networks as the scale of observation increases. Specifically, a multi-scale power-law exponent of -1.13 is fitted to

fracture and lineament lengths although we found that individually the fractures and lineaments might follow distinct power-laws rather than a common one.

## 1   Introduction

### Review

Fracture networks form the main pathways for fluid flow in fractured crystalline rocks where the matrix is largely impermeable (Nelson, 2001; Davy et al., 2006). Understanding the fluid flow in such a system is challenging, since fractures typically lack a characteristic length (Heffer and Bevan, 1990) and fractures of all sizes may contribute to the fluid flow (Davy et al.,



2006). Fracture lengths and the collective fracture network sizes span approximately 10 orders of magnitude (Marrett et al., 1999) from microfractures within individual mineral grains to continental scale tectonic structures (Bonnet et al., 2001). Char-

acteristics of the networks, such as length and orientation distributions, typically have similarities across the whole scale range (See Heffer and Bevan, 1990; Bonnet et al., 2001, and references within), and in such cases the properties are said to be scalable. However, to establish robust scaling laws that can predict fracture characteristics across multiple scales of observation, the characteristics should be investigated using a combination of multi-scale methods, such as outcrop fracture data collection and the analysis of remotely sensed lineament trace maps. This multi-scale approach will increase the overall applicability

and reduce the uncertainty associated with fracture network investigations (Bonnet et al., 2001; Bour et al., 2002; Davy et al., 2010; Bertrand et al., 2015; Heffer and Bevan, 1990; Odling, 1997; Marrett et al., 1999; Chabani et al., 2021; Palamakumbura et al., 2020). For a given fracture network, scalability may apply for all or just for some specific characteristics of the network such as lengths (e.g., Bertrand et al., 2015; Dichiarante et al., 2020) or azimuth distributions (e.g., Odling, 1997), while other properties appear scale-dependent.

Due to the unavailability of a single method that can be used to map brittle bedrock structures in all the possible scales in which they occur, multi-scale studies must use multiple methods. Modern methods of lineament and fracture interpretation used in multi-scale studies typically include outcrop-based fracture digitisation and digital elevation model and geophysics -based lineament interpretation (Bertrand et al., 2015; Hardebol et al., 2015; Dichiarante et al., 2020; Loza Espejel et al., 2020; Chabani et al., 2021; Palamakumbura et al., 2020). Multi-scale studies have earlier been mainly restricted to sedimentary rock

environments due to e.g., the significance of fracture properties on hydrocarbon exploration (Nelson, 2001). More recently, the needs of geothermal reservoir characterisation (e.g., Piipponen et al., 2022; Frey et al., 2021) and contaminant transport modelling (e.g., Hartley et al., 2018) have increased the number of studies conducted in crystalline environments (e.g., Chabani et al., 2021; Bertrand et al., 2015; Bossennec et al., 2021).

Other uncertainties within the multi-scale investigations may relate to the method, scale or the geological character of the

site. The chosen survey method within a site-specific study will limit the scale of observation, including e.g., the observed minimum and maximum fracture lengths (Bonnet et al., 2001; Heffer and Bevan, 1990), or filtering of the smallest fractures, due to the limited resolution of aerial images (Prabhakaran et al., 2019). Similar issues regarding the uncertainty occur across studies and consequently, systematic data gaps occur across fracture datasets (Marrett, 1996; Loza Espejel et al., 2020; Chabani et al., 2021). An example regarding the absence of fractures with intermediate lengths ($100 - 500\ m$) is provided from the

Discrete Fracture Network (DFN) -modelling conducted at Olkiluoto, a nuclear waste disposal facility on the west coast of Finland (Fox et al., 2012). Without published data from the intermediate length gap ($100-500\ m$) the determination of common power-law exponents for fracture and lineament length data suffers from a significant uncertainty, and the practical significance of the missing data is highlighted as fractures of these lengths are considered potentially hazardous to the integrity of spent fuel containers if earthquake-induced slip should occur along such fractures that intersect the containers at the disposal site

(Cottrell, 2022).

The character of the investigation site may affect the selection of the study method, but also cause uncertainty in the continuity and extent of observation. For field surveys conducted in areas of glacial drift, such as Finland, it is typically impossible to





map structures longer than a couple of tens of meters due to censoring by quaternary deposits and consequent lack of continuous available outcrops. Furthermore, different geological phenomena operate at different scales with different intensities, such
as glacial erosion, which preferentially erodes intensely fractured deformation zones (Glasser et al., 2020; Dühnforth et al., 2010; Skyttä et al., 2015). In contrast, polishing and abrasion dominate in more intact parts of the bedrock (Dühnforth et al., 2010; Woodard et al., 2019) where individual fractures play an insignificant role in channeling the erosion. The possibility that brittle structures from different scales have variance in their fractal nature (Davy et al., 2010) further emphasies the importance of multi-scale studies.

Although multi-scale studies are affected by the above uncertainties, the results and comparisons between different scales and used methods are useful for identifying and quantifying the uncertainties and biases that vary between scales and methods. Comparisons commonly include the analysis of geometric trace properties such as lengths, intensities and azimuths (Bertrand et al., 2015; Hardebol et al., 2015, e.g.,) but recent interest has developed towards using multi-scale data to evaluate the scalability of topological fracture network characteristics (e.g., Loza Espejel et al., 2020; Dichiarante et al., 2020) as the
topological characteristics are scale-independent by definition. The topological characteristics have direct implications on the connectivity and the fluid flow properties of the fracture network (Sanderson and Nixon, 2015, 2018) and consequently, to the DFN-modelling process (Libby et al., 2019). Although topological properties should be by definition scale-independent, recent studies (e.g., Ovaskainen, 2020; Nixon et al., 2012) have shown that the e.g., the inherent differences in source rasters used in lineament or fracture interpretation (e.g., digital elevation model for lineaments vs. RGB-image for fractures) could influence
the resulting topological network parameters.

By conducting a multi-scale lineament and fracture network investigation at Åland Islands we gain: i) A robust multi-scale dataset that adds to the currently limited pool of multi-scale studies conducted in crystalline rocks, and more specifically, bridges the gap of brittle structures with intermediate lengths (100 and 500 $m$). This specific gap in the fracture lengths has been recognised in the Olkiluoto dataset and our new results consequently lead to and reduced uncertainties in making generalised
interpolations of fracture and lineament lengths in the corresponding scale range. ii) Further insight regarding comparisons of topological network characteristics from fracture networks extracted from multiple scales which are not commonly included in multi-scale studies although they are crucial for realistic Discrete Fracture Network -modelling. Our results for e.g., the topological *Connections per Branch* parameter show a trend of increasing values as the scale of observation decreases from the outcrop fractures to lineaments.

As such, this work indicates that multi-scale, multi-method, scalability studies of the fracture networks increase the reliability of the fracture network models as compared to ones conducted in a fixed scale. Our multi-scale results enable the cross-validation of lineaments with fractures and e.g., highlight a possibility of N-S oriented lineaments to be remains of glacial flow rather than bedrock structures. Furthermore, the presented methods and results will be useful for application such as geothermal and bedrock construction projects, and they further provide a useful framework for further field analogue and characterisation
studies of local brittle structures.

We conducted mapping of brittle bedrock structures in three different scales of observation, outcrop, semi-regional and regional scale and using a combination of comparable remote-sensing methods. As the outcrop scale data (scale of observation





circa 1:10) we used fracture trace data published in Ovaskainen et al. (2022) which contains fractures with lengths from
centimeters to roughly 30 $m$. The available trace data was originally digitised from orthomosaics spanning an area of circa

20700 $m^2$. For the semi-regional scale (ca. 20 - 9000 $m$) of observations we digitised topographical lineaments from an area
of ca. 231 $km^2$ using Light Detection And Ranging (LiDAR; National Land Survey of Finland, 2010) point data, which we
further processed into a Digital Elevation Model (DEM) and visualised with multidirectional hillshading allowing optimal
recognition of topographic lineaments with any azimuth (Palmu et al., 2015). The DEM was resampled to a cell size of 5 $m$
and 150 $m$ which correspond to the semi-regional map scale of 1:20 000 and the regional map scale of 1:200 000, respectively.

The regional map scale dataset covers an area of ca. 1097 $km^2$. In the regional 1:200 000 scale, we supplemented the LiDAR
-based lineament interpretation with low-altitude airborne geophysical magnetic and electromagnetic raster data.

We characterised the fracture network properties of all three scales using geometric and topological characteristics, includ-
ing intensity (*Fracture Intensity P21* and *Dimensionless Intensity P22/B22*), azimuth, length distributions (e.g., power-law fit
attributes including exponent) and connectivity (e.g., *Connections per Trace/Branch*). We used the characterisation results to

qualitatively estimate the geological significance of the lineaments (and fractures; i.e., the likelihood that a digitised trace rep-
resents an underlying bedrock structure) of different scales and potential factors affecting the characterisation results such as
the effect of glacial erosion. Consequently, we compared the determined characteristics between all scales of observation and
specifically focused on the investigation of the potential fractal characteristics of their length distributions by determining the
possibility of modelling the trace lengths from different scales with power-law fits.

## 2   Geological Setting


The main part of Åland Islands bedrock is comprised of the 1.58 Ga Åland Batholith (Laitakari et al., 1996; Rämö and Haapala,
2005; Kosunen, 1999), which is crystalline rapakivi granite consisting mainly of Wiborgite and Pyterlite (Geological Survey of
Finland, 2017) and with an overall homogeneous character in comparison to other complex, polydeformed, crystalline bedrock
in southwestern Finland. The emplacement of the rapakivi batholiths has been generally attributed to crustal extension (Korja

and Heikkinen, 1995; Nironen, 1997), associated with an upward bulging of the mantle (Haapala and Rämö, 1992; Luosto
et al., 1990). The mesoscopic texture of the rocks is isotropic (unfoliated) as these rocks were not subjected to significant
tectonic ductile events associated with major orogenies. Some authors have associated the emplacement of rapakivi granites
with pre-existing fault and shear zones (Karell et al., 2014; Kosunen, 1999) within a strike-slip regime (Vigneresse, 2005). The
largest deformation zone within the vicinity of the Åland Batholith is the South Finland Shear Zone (Torvela and Annersten,

2005; Väisänen and Skyttä, 2007), which is a 200 km long E-W to NW-SE trending zone that experienced localised ductile
deformation at the end of the Svecofennian orogeny between 1.85-1.79 Ga (Torvela et al., 2008). The shear zone ends at the
boundary of the batholith, at least at the current erosional level (Torvela et al., 2008).

The fracture traces that represent the outcrop scale data within this study, are from the northern shoreline of Getaberget
(Figure 1), where recent contributions have revealed that the Åland Batholith was subjected to brittle faulting and generation

of associated fracture systems (Ovaskainen et al., 2022; Skyttä et al., 2022). The observed fractures comprise joints, extension





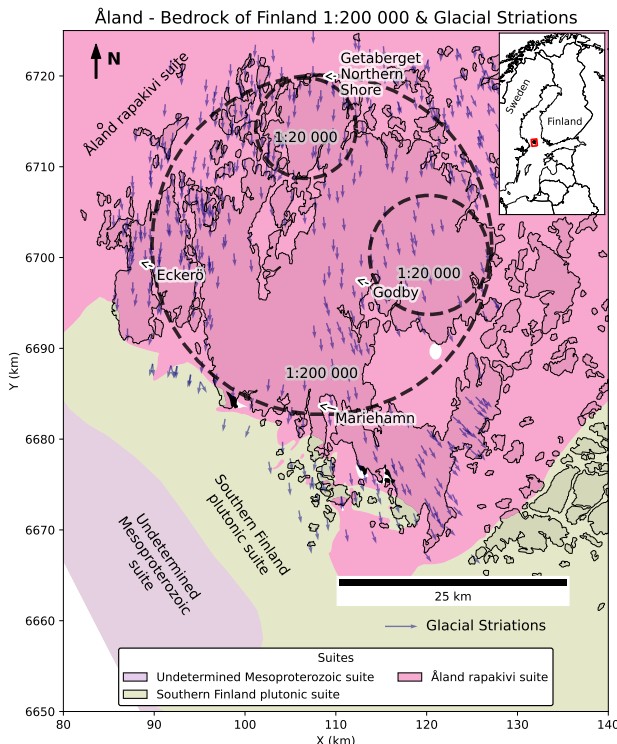

**Figure 1.** Lithological suites (Geological Survey of Finland, 2017), target areas for lineament extraction and glacial striations mapped by the Geological Survey of Finland (Geological Survey of Finland, 2014).

fractures, veins and faults, which display a range of lengths from a few $cm$ to 200 $m$. Outside larger fault zones, joints are arranged in three mutually orthogonal sets with roughly N-S and E-W sub-vertical and sub-horizontal orientation. Smaller faults are oriented mostly roughly in E-W and N-S trends but with variation. The E-W faults are parallel to subparallel with the E-W joints, but are further associated with kinematically coupled NE-SW extension fractures in their damage zones, whereas

the N-S faults have limited damage zones (Skyttä et al., 2022).

The topography and quaternary deposits on the Åland islands are shaped by several glaciation cycles during the Pleistocene. Glacial striations (Figure 1) and distinct glacial landforms such as flutings, typically visible in digital elevation models (E.g., Ojala and Sarala, 2017), indicate that the glacier moved in approximately N-S directions during the latest glacial periods. Besides the smooth abrasion-related glacial erosion (see above) the fracture systems within the bedrock contributed towards

glacial quarrying, which was particularly intense within individual larger faults (Skyttä et al., 2022).





**Table 1.** Definitions of scales of observation.

| Representative Factor / Name | Cell Size [$m$] | Total Target Area [$m^2$] |
|---|---|---|
| 1:10 | 0.0055 | 20,708 |
| 1:20 000 | 5.0000 | 230,726,255 |
| 1:200 000 | 150.0000 | 1,096,918,465 |

## 3   Data & Methods

We identify the different scales of observation used for fracture and lineament interpretation by the representative factor i.e., the ratio between a distance on a "map" and the distance on the ground (Goodchild, 2011). E.g., 1:10 states that 1 meter on the "map" represents 10 meters in nature. However, the usage of the representative factor for representing the scale of digital data displayed on a computer screen is not well defined due to e.g., differences in software and display hardware (Goodchild, 2011). To better specify the scale of observation the use of areal extent and resolution of data are preferred (Goodchild, 2011, 2001) and we display these characteristics in Table 1. The used representative factors should only be considered a convenient naming schema as the resolution better defines the scale of observation. Generation of traces at all the involved scales is conducted remotely from aerial datasets, which allows comparisons between the well-represented sub-vertical features, while the sub-horizontal ones are underrepresented and hence not further discussed in this paper.

### 3.1   Data

Brittle bedrock discontinuities can be classified based on several characteristics including fracture filling, kinematics and geometry, resulting in a number of terms that can be used to refer to the different types (e.g., joint, vein, fault and fracture; Odling et al., 1999). We use the most general term fracture when referring to brittle discontinuities in general, as we do not discriminate between different types in the analysis. Categorisation of the outcrop fractures digitised from orthomosaics could be done in the field but to gather representative data on circa 40000 fractures would require significant time-investment. In addition, the results would be difficult to integrate with the remotely digitised data as the scale of observation would likely be different. Furthermore, field verification of lineaments is much more difficult due to quaternary cover and preferential erosion of the depressions. We consequently attempt to analyse the data without specific prior knowledge of the types of features the fractures and lineaments represent. We refer to the networks of both fractures and lineaments as fracture networks and use it as a general term for the collections of fracture or lineament traces.

We used existing fracture trace data published by (Ovaskainen et al., 2022) from the northern shore of Åland Islands as the outcrop scale 1:10 dataset (Figure 2). The fracture data was collected from 2D raster orthophotos from within 13 circular target areas along the E-W trending Getaberget shoreline. The circle diameters ranged from 20 to 50 meters and the number of digitised traces within the circles varied from 358 to 7319. The dataset requires no modifications for the purposes of this study. However, rather than investigating each target area individually we merge the trace data (n=42499) into a single dataset of traces







**Figure 2.** A. Overview of the Getaberget outcrop with local lithology (Geological Survey of Finland, 2017). Figure from Ovaskainen et al. (2022). B. Drone imaged orthomosaics superpositioned with fracture digitisation target areas and digitised fracture traces. Data from Ovaskainen et al. (2022).

that are cropped specifically to the associated target areas resulting in a trace count of 41544. There are significant variations in fracture network properties between the target areas but without apparent spatial trends that could be used to correlate the characteristics with their location (Ovaskainen et al., 2022). Therefore, the merging of the data produces an aggregated dataset

of the fracture characteristics, representative for the entire Getaberget shoreline study area.





Lineaments in this paper are defined as sub-linear lines on the surface of the Earth (See e.g., Tyrén, 2011; Nur, 1982) which are visible in one or more datasets such as in a digital elevation model or in a geophysical raster. All lineaments digitised for this paper are interpreted remotely by three operators working collaboratively and cross-verification of interpretations between operators was done to try to minimize subjective bias (See e.g., Andrews et al., 2019; Bond et al., 2007, for further discussion

on subjective biases). The lineaments have not been geologically verified in the field. We used the publicly available airborne LiDAR point data published by the National Land Survey of Finland (2010) to create a DEM for the purposes of lineament interpretation in the 1:200 000 and 1:20 000 scales. The used point data has a point cloud density of 0.5 $points/m^2$ and the mean altitude error is 0.3 $m$. Specifically, a cell size of 150 $m$ is used for the 1:200 000 scale and 5 $m$ is used for the 1:20 000 scale interpretation. We visualize the DEM using a multi-directional oblique hillshade on top of the DEM raster to

highlight the topographical valleys and slopes (Palmu et al., 2015). The hillshade has a z-factor of 1, the used altitude of light is 45 degrees and illumination azimuths are 225, 270, 315 and 360 in degrees. We overlaid the transparent (alpha value 0.3) white-to-black hillshade upon the blue-to-red DEM raster to allow the optimal recognition of linear structures with variable trends. Furthermore, we calculated the color-scales of both rasters from the current extent of the canvas i.e., the coloring is recalculated dynamically as the interpreter pans or zooms the map.

In addition to the LiDAR DEM topographical raster, we interpreted geophysical lineaments in the 1:200 000 scale using regional low-altitude magnetic and electromagnetic aerogeophysical rasters (Hautaniemi et al., 2005). Flight-altitude and flight line spacing during the acquisition of magnetic data were 30 $m$ and 200 $m$, respectively, and the acquired raw data was further processed into various rasters with a 50 $m$ cell size. We resampled all rasters we used for the 1:200 00 scale interpretation to a cell size of 150 $m$ to match the resolution of the also resampled 1:200 000 scale LiDAR DEM.

We used three magnetic rasters: i) total field DGRF-65 greyscale, ii) sharp-filtered total field DGRF-65 grayscale and iii) tilt derivative (Verduzco et al., 2004). Based on these three magnetic raster maps, we interpret lineaments along the recognised linear magnetic maxima and minima, which ideally correlate with deformation zones characterised by metamorphically generated magnetite or pyrrhotite, or fluid-induced alteration and leeching, respectively (See Paananen and Posiva Oy, 2013; Middleton et al., 2015, and references within both).

We used one electromagnetic raster from the same national surveying program, a 3 $kHz$ quadrature component grayscale map, which we used to interpret electromagnetic lineaments. Lineaments from this map are interpreted along the local minima which correspond to either i) electrically conductive brittle damage zones (with water and/or conductive minerals) or ii) linear topograhic depressions caused by the preferential erosion of brittle damage zones, and containing conductive soils with clay minerals and peat alongside rainwater (See Paananen and Posiva Oy, 2013; Middleton et al., 2015, and references within both).

After interpretation of lineaments from each source (the LiDAR DEM, the magnetic maps and the electromagnetic map) we integrated the lineaments into a single dataset where lineaments interpreted from different sources were merged based on their superposition. Overlapping lineaments were merged along the overlapping parts while the deviating segments such as splays were preserved. This integrated lineament dataset is the representative dataset used for the 1:200 000 scale in all analyses. We use QGIS 3.14 (QGIS Development Team 2020) and ARCMAP 10.6.1 to digitize the lineaments as georeferenced polylines.

Similar to Ovaskainen et al. (2022), we used the snapping functionality present in both software packages in order to honor the



true abutment relationship between the traces, and consequently, document realistic topological relationships of the network (Nyberg et al., 2018). To verify the topological consistency of the lineaments the traces are validated with a Python package, FRACTOPO, which provides a validation utility to find e.g., V-nodes and overlapping lineament sections (Ovaskainen, 2022b). The 1:200 000 scale lineaments were digitised by three persons, including the main author, while the 1:20 000 scale lineaments

were digitised solely by the main author. The interpretations were done in circular target areas to remove the uncertainty related to the shape of the interpretation area (Mauldon et al., 2001; **?**; Ovaskainen et al., 2022).

### 3.2 Lineament and Fracture Network characterisation and Comparison

The interpreted lineament dataset is comparable to the used fracture trace data because it has been digitised and validated similarly and is therefore analysable using the same tool used by Ovaskainen et al. (2022), FRACTOPO. All functionality

required for the multi-scale analysis of lineament and fracture trace data of this paper are implemented in the main FRACTOPO software repository (Ovaskainen, 2022b). However, to allow easier reproducibility of the more specific analysis, the results of this study and the explicit FRACTOPO-based workflow, analyses and figures are presented in a separate open repository (Ovaskainen, 2022a).

For each scale of observation, 1:10, 1:20 000 and 1:200 000, we present a set of network characterisation results. *Fracture*

*Intensity P21* is calculated from the total trace length occurring within an area. The derivatives of it, *Dimensionless Intensity P21 and B22*, are calculated by multiplying the value of *Fracture Intensity P21* by the characteristic trace or branch length, respectively (Sanderson and Nixon, 2015). As these two parameters have no units, i.e., they are dimensionless, they are well suited for intensity comparisons between scales. We used equal-area length weighted rose plots to visualize the azimuth distributions (Ovaskainen et al., 2022; Sanderson and Peacock, 2020) and further subdivided them into sets that occur in all or in at

least two of the scales. To analyse network topology and to present topological network characteristics, we determined the topological branches and nodes (Manzocchi, 2002; Mäkel, 2007; Sanderson and Nixon, 2015; Nyberg et al., 2018) of the network using FRACTOPO. Nodes represent interactions between traces or trace abutments in isolation. Specifically, Y-nodes represent trace abutments to each other, X-nodes represent traces cutting through each other and I-nodes represent trace abutments in isolation (Manzocchi, 2002; Mäkel, 2007; Sanderson and Nixon, 2015). The node types can be generalised to be connected or

unconnected where the X- and Y-nodes are connected (C) and I-nodes unconnected (I). Using this generalisation the branches, which are the trace segments between the nodes, can be given types of C-C, C-I and I-I where the type is determined by the end nodes of each segment (Sanderson and Nixon, 2015). The branches and nodes were analysed for scale-independent estimates of network connectivity by plotting the relative proportions of different types of nodes and branches into ternary plots (Manzocchi, 2002; Sanderson and Nixon, 2015) and by calculating parameters *Connections per Trace* and *Connections per Branch*

(Sanderson and Nixon, 2015).

Regarding the fracture length, we determined power-law, lognormal and exponential distribution fits to the trace length data using FRACTOPO, which in turn uses the POWERLAW-package (Alstott et al., 2014) for Maximum Likelihood Estimation of the fits following Clauset et al. (2009). Following Bonnet et al. (2001) and Clauset et al. (2009), the power-law modelled distribution of lengths $n(l)$ is represented as a function of the power-law exponent $a$ and a constant $A$:





$$n(l) = A \times l^a$$

Along with the length distribution fits, POWERLAW-package automatically determines the cut-off value for the length data below which lengths do not seemingly fit the same power-law exponent. The cut-off at the tail end of the distribution is attributed to the fixed scale of observation (See e.g., Bonnet et al., 2001; Pickering et al., 1995, for discussion on truncation and censoring sampling issues for fractures). For fracture trace data, the need for a truncation cut-off is attributed to the insufficient ability to digitize the smallest fractures visible in the images due to insufficient resolution (Pickering et al., 1995; Bonnet et al.,

2001). To visualize the length distributions, we plotted the lengths on the x-axis and complementary cumulative number of the length distribution on the y-axis. Both axes, x and y, are logarithmically scaled. Cumulative number in this study means a running integer number starting from 1 (the shortest fracture), then counting upwards and ending at the longest. The prefix, complementary, means that the cumulative number is then inversed so that the longest fracture has the smallest value. If the data is power-law distributed the scatter data on the plot will follow a sub-linear trend with an expected deviation from the

trend at some cut-off value. We tested the goodness-of-fit of a power-law trend by comparing the fit to the fit of a lognormal distribution. We display the loglikelihood ratio $R$ and ratio significance $p$ values of the likelihood comparisons (Alstott et al., 2014). The loglikelihood ratio $R$ is positive when the power-law trend is more likely and negative when the lognormal trend is more likely. High statistical significance of the comparison is described by low $p$ values, where a $p$ value of less than $0.1$ is considered statistically very significant (Clauset et al., 2009). Because the power-law fit typically requires a cut-off when

comparing the different distributions, all comparisons are made to the cut-off truncated data rather than the full length data to enable the comparison of the fits as recommended by Clauset et al. (2009). Furthermore, we analysed the lengths of the network branches and use the same determination method as used for the traces to fit different potential distributions to the branch length data. The lengths of topological branches are less subject to subjective bias related to the interpreter (Sanderson and Nixon, 2015; Loza Espejel et al., 2020; Sanderson and Nixon, 2018). Consequently, the results of length distribution

analysis of branches are potentially better suited for comparisons between scales of observation in this study or in comparisons to other studies of branch length distributions (Sanderson and Nixon, 2015; Loza Espejel et al., 2020; Sanderson and Nixon, 2018; Lahiri, 2021).

       As we had trace length data of structures from multiple scales of observation, we could investigate the potential fractal nature of the lengths by plotting all trace length data onto a single plot and fitting a power-law function to all data or to data truncated

by individual cut-offs (Sornette et al., 1990; Davy, 1993; Bonnet et al., 2001; Davy et al., 2010). We conducted this analysis as there is physical rationale for brittle structure trace lengths to follow power-law distributions across different scales of observation (Bonnet et al., 2001). To normalize the scale of observation we divided the Complementary Cumulative Numbers (CCM) of each scale dataset by the total area of the target area to get the Area-Normalised Complementary Cumulative Numbers (ANCCM) following Bonnet et al. (2001). Rather than using all trace length data, we used the aforementioned cut-

offs determined for individual length distributions to remove the tails (lowest trace lengths) from the distributions before fitting the multi-scale trend. To fit the trend we could not use the POWERLAW-package as it does not support automatic fitting to multiple, separate, distributions simultaneously. Rather, we used a least squares polynomial fit function, POLYFIT from the





**Table 2.** Counts of digitized fractures and lineaments from each source that intersect their respective target areas.

| Raster Source | Count |
| --- | --- |
| LiDAR 1:200 000 | 150 |
| Magnetic 1:200 000 | 48 |
| Electromagnetic 1:200 000 | 21 |
| Integrated 1:200 000 | 201 |
| LiDAR 1:20 000 | 609 |
| Orthomosaics 1:10 | 41544 |

NUMPY Python package (Harris et al., 2020), and assessed the multi-scale goodness-of-fit with Mean Squared Logarithmic Error (MSLE). The fit is done to the logarithm of the length and CCM data as implemented in FRACTOPO (Ovaskainen,
2022b). Using multi-scale azimuth sets determined from a visual inspection of rose plots of the scales of observation, we could further investigate the possibility of fitting multi-scale power-law trends to multi-scale length data that is categorised by azimuth set. The approach has the potential to reveal differences between length distributions of fractures and lineaments in different azimuths (E.g., Skyttä et al., 2021; Ceccato et al., 2022). Of particular interest is whether the effect of glacial erosion has caused differences in the length distributions of features in different azimuths.

**4 Results**

**4.1 Lineament Interpretation**

Lineament interpretation and subsequent integration of topographic and geophysical lineaments in the Åland Islands resulted in 201 integrated lineaments in the 1:200 000 scale. The number of lineaments from each different interpretation source in the 1:200 000 scale is displayed in Table 2. The addition of geophysical rasters to complement LiDAR DEM -based interpretation
resulted in a significant number of additional lineaments. Specifically, a significant number of geophysical linemeants with a NW-SE trending azimuth were added (Figure 3). The effect of glacial erosion, in the N-S direction, is apparent in the LiDAR raster but is not visible in any of the geophysical rasters. Practically no N-S oriented lineaments were either interpreted from the geophysical rasters, whereas in the LiDAR raster a significant number of such lineaments were digitised in the 1:200 000 scale. Lineament digitisation in the scale 1:20 000 was limited to the LiDAR DEM raster as the geophysical rasters lacked the
resolution for the more accurate extraction possible in the 1:20 000 scale. The digitisation resulted in 609 lineaments which are visualised in Figure 4A. The northern target area for 1:20 000 lineament interpretation covers the Getaberget hill area and the surrounding terrain (Figure 1). Because the hill area is distinctly better exposed than the neighbouring terrain, the interpreted lineament density is higher in this better exposed and elevated area (Figure 4A).



**Figure 3.** All subfigures (A, B and C) contain the 1:200 000 scale lineaments that were interpreted using the displayed raster. In the case of subfigure B two other magnetic maps were used (Total field DGRF-65 and Tilt derivative DGRF-65) to interpret the displayed lineaments (See Appendix A). A. Map of a Light Detection And Ranging (LiDAR) -based digital elevation model (DEM) with a hillshade overlay. B. Map with grayscale visualised sharp-filtered DGRF-65 magnetic data. C. Map with grayscale visualised 3 $kHz$ quadrature component electromagnetic data. D. Same map as subfigure A. but with the integrated lineaments overlaid on top.

## 4.2 Multi-scale Network Characterisation

Scalar network characteristics of each scale dataset are collected in Table 3. The characterisation consists of geometric and topological parameters which can be used to compare the scales. Of special interest are the dimensionless parameters (*Dimen-*




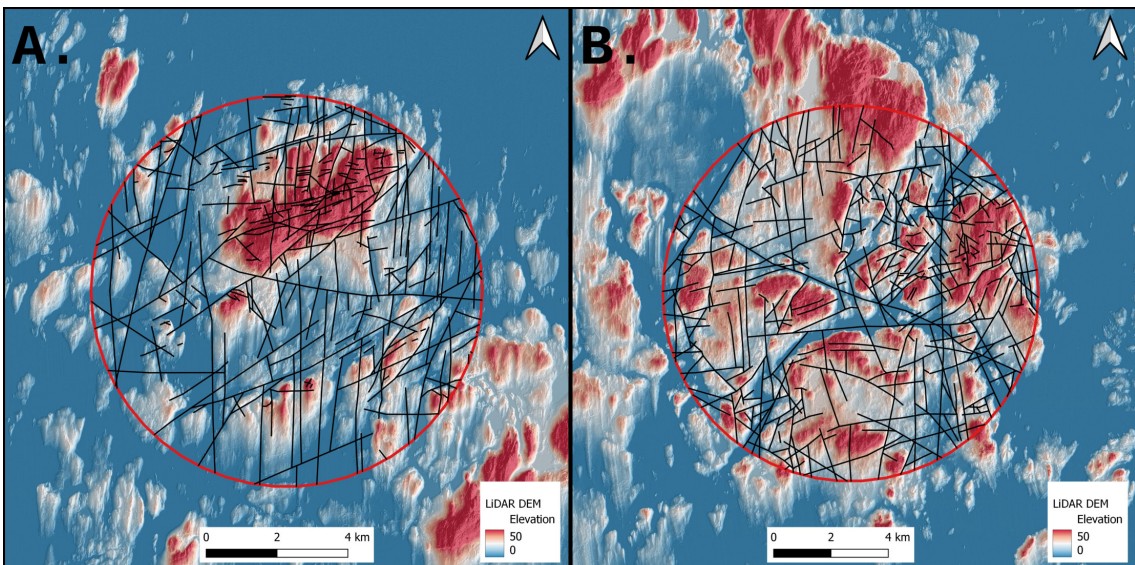

**Figure 4.** LiDAR DEM overlaid with digitised 1:20 000 scale lineaments and the two separate target areas. A. Northern area of Getaberget. B. Eastern area near Godby.

*sionless Intensity P22 and B22*, *Connections per Trace*, *Connections per Branch* and *Trace and Branch Power-law Exponents*) as these are especially suited for comparisons between scales of observation (Sanderson and Nixon, 2015; Goodchild, 2001). The scale-dependant *Fracture Intensity P21* has an expected trend of higher intensity with higher scale with the 1:10 scale

having the highest value and 1:200 000 the lowest. Same trend is seen with *Dimensionless Intensity B22*. *Connections per Trace* and *Connections per Branch* display a trend with values decreasing as the scale increases with the 1:10 scale having the lowest value.

The individual azimuth and length analysis results for each scale are visualised in Figure 5 where trace azimuths are represented with equal-area length-weighted rose plots (Sanderson and Peacock, 2020) and trace and branch lengths are modelled

with power-law, lognormal and exponential fits. Based on the displayed rose plots (Figure 5A), three distinct azimuth sets occur in all scale datasets (Table 4). The sets occur with different intensities in different scales, which is recorded in Table 4 with a numbering: 1 equals the most abundant set and 3 the least abundant. Relative abundance is based on the displayed percentages of total trace length of each set in Figure 5A. The relative abundance of the sets differs greatly between the scales and when the set is labeled as the least abundant (3) the occurrence of it in the scale is vague. E.g., the N-S set is barely visible in the 1:10

scale rose plot (Figure 5A) with only a minor local maximum detectable at around 175 degrees. Similarly, the WNW-ESE set is barely detectable in the 1:200 00 scale rose plot without any detectable local maximum.

The exponents of the fitted power-law trends for trace lengths vary drastically when comparing fractures and lineament scales: The 1:10 and 1:20 000 scale fracture traces have fitted *Trace Power-law Exponents* of -2.095 and -2.259, respectively, whereas the 1:200 000 traces have an exponent of -1.14 (Figure 5B) . However, 1:10 and 1:200 000 scale branch lengths have



**Table 3.** Basic network descriptions of all scales of observation with units displayed when applicable. EM = Electromagnetic 3kHz quadrature. Mag = Magnetic rasters. [a] Based on node counting (Sanderson and Nixon, 2015).

| Name | 1:10 | 1:20 000 | 1:200 000 |
|---|---|---|---|
| **Data Source(s)** | Orthomosaics | LiDAR | LiDAR+EM+Mag |
| **Number of Traces** [a] | 40654 | 539 | 157 |
| **Number of Branches** [a] | 93125 | 1965 | 1151 |
| **Number of Traces (Real)** | 41618 | 621 | 206 |
| **Area** $[m^2]$ | 20707.55 | 2.31e+08 | 1.10e+09 |
| **Trace Max Length** $[m]$ | 34.86 | 13223.49 | 32635.24 |
| **Trace Mean Length** $[m]$ | 1.0 | 1290.21 | 6698.69 |
| **Branch Max Length** $[m]$ | 9.57 | 2789.28 | 8501.36 |
| **Branch Mean Length** $[m]$ | 0.45 | 407.75 | 1198.38 |
| **Fracture Intensity P21** $[\frac{m}{m^2}]$ | 2.01 | 3.47e-03 | 1.26e-03 |
| **Dimensionless Intensity P22** | 2.01 | 4.48 | 8.43 |
| **Dimensionless Intensity B22** | 0.9 | 1.42 | 1.51 |
| **Trace Power-law Exponent** | -2.09 | -2.26 | -1.14 |
| **Branch Power-law Exponent** | -3.37 | -2.47 | -2.96 |
| **X** | 9771 | 419 | 423 |
| **Y** | 32929 | 588 | 148 |
| **I** | 48380 | 490 | 167 |
| **C - C** | 52586 | 1460 | 946 |
| **C - I** | 31343 | 391 | 160 |
| **I - I** | 8311 | 44 | 2 |
| **Connections per Trace** | 2.1 | 3.74 | 7.25 |
| **Connections per Branch** | 1.48 | 1.75 | 1.85 |

**Table 4.** Visually determined multi-scale trace azimuth sets along with relative abundance in each scale where 1 equals the most abundant of the sets and 3 the least abundant.

| | Relative Abundance | | |
|---|---|---|---|
| Azimuth Set Label and Range (degrees) | 1:10 | 1:20 000 | 1:200 000 |
| N-S (155-25) | 3 | 1 | 2 |
| NE-SW (25-75) | 2 | 2 | 3 |
| WNW-ESE (85-135) | 1 | 3 | 1 |





**Figure 5.** A. Equal-area length-weighted rose plots of trace azimuths of each scale along with the percentage of total length that each set contains. The determined sets do not cover all azimuths and therefore the percentages do not add up to 100 %. B. Length distributions of traces of each scale on plots with complementary cumulative number (CCM) on the y axis and trace length on the x axis. The distributions are fitted with power-law, lognormal and exponential fits and the automatically determined power-law cut-off is indicated with the vertical dashed line and text. C. Length distributions of branches with the same setup as subfigure B except for the x axis which has branch lengths rather than trace lengths.

relatively similar exponents of -3.37 and -2.96, respectively, whereas the 1:20 000 scale branch lengths have an exponent of -2.47. Further characterisation of the trace length distributions is displayed in Table 5 where the power-law fit is compared to





the lognormal fit (See rows with "All") and cut-off proportions are displayed. Comparisons to exponential fits are not displayed as even visual inspection shows that it does not model the lengths well (Figure 5B). For all scales the lognormal fit is more probable according to the $R$ values. However, based on the $p$ values the lognormal fit is significantly more probable than the

power-law fit for the 1:200 000 scale ($p$ value less than 0.1) while power-law remains a possible alternative for both the 1:10 and 1:20 000 scale datasets. The cut-off proportion (i.e., amount of data removed by the application of the cut-off) for the 1:10 scale is very high with 97.82 % of data being cut off. The proportion is similarly high for the 1:20 000 traces with a value of 88.08 % and significantly lower for the 1:200 000 traces with a value of 35.92 %.

Table 5 also contains fit results to azimuth set-wise categorised trace lengths for each scale. For all scales the set-wise fits

do not drastically differ from the fits to all traces in terms of power-law exponents. The power-law fits for 1:10 scale azimuth set lengths remain candidate fits expect for the WNW-ESE-set where the lognormal fit is significantly more probable with a $p$-value of $1.45e-05$. For both the 1:20 000 and 1:200 000 scales all azimuth set-wise fits have $p$ values of over 0.1 indicating that the power-law fit cannot be ruled out as a candidate model for the trace lengths of each set. However, for these comparisons, it should be kept in mind that the azimuth sets occur with very different intensities across the scales of observation (Table 4).

Furthermore, as the traces are subdivided into sets, the sample count within each set decreases which lowers the reliability of the results especially for the lineament datasets which have lower sample counts (Table 4). Regardless of these uncertainties, a common trend is also noticeable where the WNW-ESE set traces have the lowest power-law exponents in all scales.

The multi-scale power-law fit to all traces is visualised in Figure 6A along with fits to trace lengths categorised by the previously determined azimuth sets in Figure 6B-C Based on visual inspection of the plot with all trace data (Figure 6A)

the 1:10 scale fractures and the 1:200 000 scale lineaments seem to follow a common power-law trend while the 1:20 000 lineaments deviate from it. However, the tail cut-offs majorly affect the distributions and the resulting fits. Also, the largest length traces (head) within each scale deviate from the common trend. When comparing the azimuth set categorised multi-scale length distributions (Figure 6B-C), the WNW-ESE set is somewhat anomalous compared to the rest. Due to having lower cut-offs, determined from individual distributions (Table 5), a higher proportion of length data is used with the WNW-ESE

data which increases the Mean Squared Logarithmic Error (MSLE) but, based on visual inspection, a common trend seems more likely for 1:10 scale fractures with intermediate lengths (length data around the cut-off value of 1.75 $m$) rather than the higher length fractures at the head of the distribution. The exponent values of the power-law trends are quite similar across the different arrangements (Figure 6). The NE-SW set has the highest exponent of -1.12, closely followed by the exponent of all traces (Figure 6A-D) and the exponent of the N-S set trace lengths (-1.19). The WNW-ESE trending traces have the lowest

exponent of -1.30 which slightly deviates from the other exponents (Figure 6D). The trend of the individual 1:10 fracture and 1:20 000 lineament distributions in Figures 6A and 6B seem to indicate a power-law exponent lower than the trends of the lineaments as is also evidenced by fits to the individual distributions where the exponent is -2.095 for the 1:10 scale fractures and -2.26 for the 1:20 000 scale lineaments (Figure 5).

The proportions of topological node types (X, Y and I) and branch types (C-C, C-I, I-I) from Table 3 are visualised in

Figure 7 with ternary plots (Manzocchi, 2002; Mäkel, 2007; Sanderson and Nixon, 2015). From both the node ternary plot (Figure 7A) and branch ternary plot (Figure 7B) a trend can be observed where the apparent connectivity of the network





**Table 5.** Parameters of length distribution fits for traces and branches for all scales along with set-wise fits of traces for all scales. PL = power-law, LN = lognormal. R-value is the loglikelihood ratio where a positive value indicates that the power-law fit is more likely and a negative value that the lognormal fit is more likely. The p-value represents the significance of the likelihood where low values (<0.1) correspond to high statistical significance.

| Name | n | PL Exp. | PL Cut-Off [$m$] | Cut-Off % | LN Sigma | LN Mu | PL vs. LN R | PL vs. LN p |
|---|---|---|---|---|---|---|---|---|
| 1:10 Traces All | 41618 | -2.09 | 5.87 | 97.82 | 1.5 | -2.07 | -1.48 | 0.14 |
| 1:10 Branches All | 94014 | -3.37 | 2.77 | 99.11 | 0.95 | -1.46 | -1.41 | 0.16 |
| 1:10 Traces N-S (155-25) | 9266 | -1.81 | 3.53 | 94.53 | 2.42 | -8.28 | -0.61 | 0.54 |
| 1:10 Traces NE-SW (25-75) | 11220 | -2.34 | 5.13 | 97.55 | 1.48 | -2.68 | -0.61 | 0.54 |
| 1:10 Traces WNW-ESE (85-135) | 16138 | -1.56 | 1.75 | 82.95 | 1.45 | -1.58 | -4.34 | 1.45e-05 |
| 1:20 000 Traces All | 621 | -2.26 | 3544.37 | 88.08 | 0.86 | 7.27 | -0.96 | 0.34 |
| 1:20 000 Branches All | 2037 | -2.47 | 683.96 | 83.26 | 0.67 | 6.09 | -2.59 | 9.71e-03 |
| 1:20 000 Traces N-S (155-25) | 199 | -2.33 | 3451.3 | 79.9 | 0.73 | 7.64 | -0.86 | 0.39 |
| 1:20 000 Traces NE-SW (25-75) | 180 | -1.56 | 1483.87 | 66.67 | 1.88 | 2.97 | -0.47 | 0.64 |
| 1:20 000 Traces WNW-ESE (85-135) | 142 | -1.35 | 1046.52 | 66.2 | 1.54 | 5.04 | -0.67 | 0.5 |
| 1:200 000 Traces All | 206 | -1.14 | 4178.75 | 35.92 | 1.35 | 7.7 | -1.8 | 0.07 |
| 1:200 000 Branches All | 1195 | -2.96 | 2382.63 | 87.11 | 0.57 | 7.39 | -1.65 | 0.1 |
| 1:200 000 Traces N-S (155-25) | 72 | -1.22 | 3854.44 | 34.72 | 1.44 | 7.1 | -0.9 | 0.37 |
| 1:200 000 Traces NE-SW (25-75) | 56 | -1.2 | 5621.01 | 50.0 | 1.12 | 8.46 | -1.03 | 0.3 |
| 1:200 000 Traces WNW-ESE (85-135) | 63 | -1.14 | 4692.67 | 31.75 | 1.55 | 7.19 | -0.84 | 0.4 |

increases as the scale of observation becomes smaller i.e., the resolution used for interpretation is poorer. Specifically, the proportion of X-nodes increases as the scale becomes smaller and is the highest for the scale 1:200 000 network. The trend is also observable from the values of *Connections per Branch* and *Connections per Trace* (Table 3).

# 5  Discussion

## 5.1  Gap Between Outcrop and Lineament Data

Our use of the 1:20 000 scale of observation in digitising topographical lineaments has the potential to reduce the uncertainty related to the brittle structures of intermediate length (100-500 $m$) which are commonly missing from studies based on outcrop digitisation and lineament interpretations (Marrett et al., 1999; Strijker et al., 2012; Loza Espejel et al., 2020; Fox et al., 2012). This missing length data was found to be a problem during the creation of a DFN-model for the Olkiluoto spent fuel disposal facility where fracture or lineament data could not be empirically collected with these lengths (Fox et al., 2012).






**Figure 6.** A. Multi-scale power-law fit to all trace length data. B-D Fits to trace length data categorised by defined azimuth sets (Table 4).

However, the DFN-model required the creation of fractures of all sizes using a common scaling law (or laws), including ones in this missing length range. Without empirically detected fractures within this size range, an uncertainty remained on the validity of generating fractures of these sizes. The generation was done using a length distribution model derived from outcrop fracture data or alternatively from lineament data (or both). All three options required the extrapolation or interpolation into the unknown intermediate length range (Fox et al., 2012). In our study, although we produce lineament length data in the 100-500 $m$ length range (Figure 5B), the optimisation of the power-law fit to all of the 1:20 000 scale length data resulted in a cut-off of 3544 $m$ (Figure 5; Table 5). This cut-off can be estimated to be the lowest length lineaments which we can





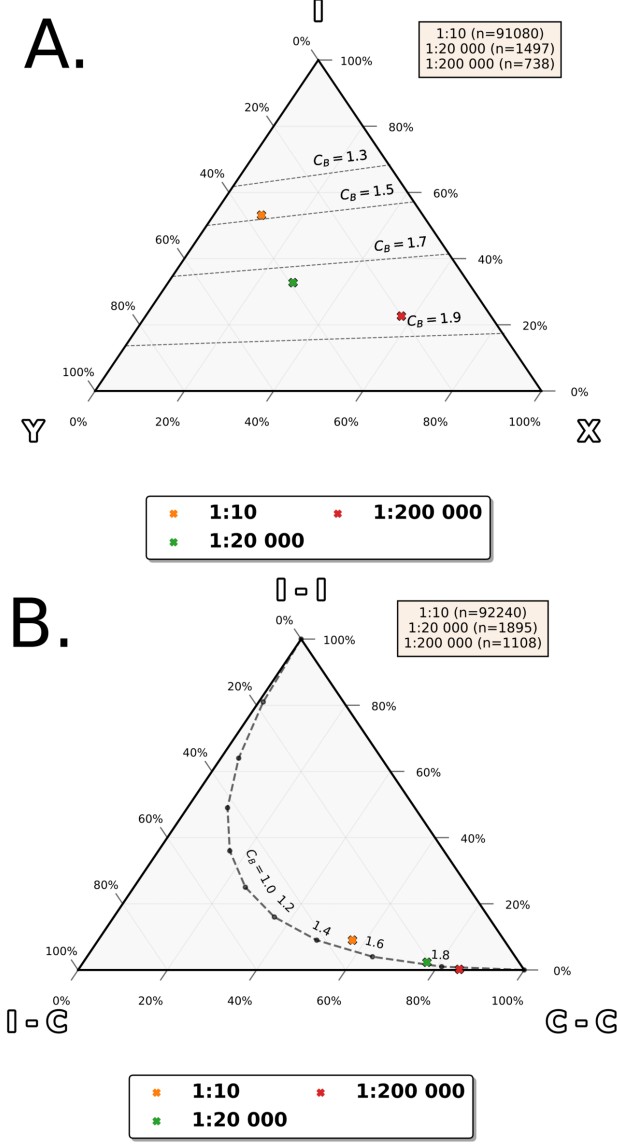

**Figure 7.** A. Ternary plot of topological XYI node proportions for each scale. B. Ternary plot of topological (C-C, C-I and I-I) branch proportions.

consistently interpret without truncation effects caused by resolution of the LiDAR DEM, assuming that the lineament trace

lengths follow a power-law. Consequently, a significant length data gap still occurs between the 1:10 outcrop scale and the 1:20 000 scale lineament scale (Figure 5). The resolution of the LiDAR DEM could enable the interpretation of lineaments within this length gap by using a larger scale of observation (e.g., 1:10 000). However, in the vast majority of the 1:20 000 area, based on visual observation of the LiDAR DEM (Figures 3 and 4), the landforms would become less sub-linear and more uncertain,



with regard to if they reflect the structures of the underlying bedrock. In contrast, where quaternary deposits do not overlay
the bedrock, such as at the Getaberget shoreline outcrops, the bedrock features are directly observable from the DEM. These
areas are however limited when considering both their shape and areal extent in comparison to the low resolution of DEM and
are better surveyed with drone photography. Digitisation of fractures longer than the diameter of the circular target areas used
at Getaberget (50 $m$) could be possible but they could not be sampled using circular target areas as the width of the polished
part of the outcrops is limited to not much higher than the diameter of 50 $m$ (Ovaskainen et al., 2022). Using non-circular,
irregularly, shaped target areas would add uncertainty to the orientation distributions we sample from the target area which
would, consequently, decrease the significance of the results (Mauldon et al., 2001; Rohrbaugh et al., 2002; Ovaskainen et al.,
2022).

## 5.2 Factors Affecting Analysis

A distinct difference in the data between the three scales, is the significantly higher number of traces within the 1:10 scale
compared to either of the lineament trace datasets (Table 3). Previous studies on the representative trace count required for
trace length distributions analysis have suggested minimum trace counts of 150 to 300 (Priest, 1993), and below (Bonnet
et al., 2001; Zeeb et al., 2013). The 1:200 000 scale lineament trace count of 201 (Table 5) is lower than the upper threshold
by (Priest, 1993) but higher than the minimum recommendations of 200 and 110 by Bonnet et al. (2001) and Zeeb et al.
(2013), respectively. The study by Ovaskainen et al. (2022) on the Getaberget trace dataset, which we use as the 1:10 scale
data, suggested that the sample area (and simultaneously trace count) could be significantly reduced to still result in the same
characterisation results for the Getaberget area. But we cannot rule out the possible effect of low lineament trace counts on the
analysis of trace lengths for the 1:200 000 scale. The effect of lower count of lineaments might also cause the subjective bias
related to the interpreters of the lineaments to have more effect as each individual choice in the lineament interpretation has
more weight.

The lineament interpretation in scale 1:20 000 is solely based on the LiDAR DEM whereas interpretation in scale 1:200
000 uses geophysical rasters in addition to the DEM to enhance the detection of bedrock structures. This could cause the
interpreted structures to differ between the scales where the structures with high geophysical and but low topographical signals
would be more likely detected in the 1:200 000 scale interpretation. This could explain the lack of a detectable WNW-ESE
set in the 1:20 000 scale lineaments (Figure 5; Table 4) which is detected in the geophysical rasters (albeit as mostly NW-
SE trending lineaments) in the 1:200 000 scale (Figure 3) and corresponds, azimuth-wise, to the major South Finland Shear
Zone (SFSZ) that seemingly abuts next to the main Åland island (Torvela et al., 2008). The WNW-ESE set is represented
by a single long lineament that cuts through the entire eastern 1:20 000 target area (Figure 3B). However, because it cuts the
target area from both ends, it is removed from any further analysis due to the boundary weighting methodology implemented
in FRACTOPO (Figure 5 by Ovaskainen et al., 2022). The set is otherwise represented only by very few small lineaments
(Figure 5A). The similar azimuth trend with the SFSZ could indicate that the structures of the Åland rapakivi batholith might
inherit a structural trend from the SFSZ which would, consequently, increase the geological significance of the WNW-ESE
oriented lineaments. The set is more detectable in the 1:10 scale fracture traces of which some correspond to field surveyed





faults (Skyttä et al., 2022; Ovaskainen et al., 2022). Based on these observations, we suspect that the lack of geophysical data in the scale 1:20 000 could result in a lack of structures that have relatively low topographical signals in that scale with only
the largest WNW-ESE structures being detectable (Figure 4B). Consequently, we recommend the supplementation of the 1:20 000 scale interpretation with high resolution geophysical data as it would increase the certainty of lineament interpretation in that scale. These uncertainties limit the possibility of using the full resolution of the LiDAR DEM to map lineaments with intermediary lengths (100-500 $m$; Continued discussion from Section 5.1). However, geological differences in the quaternary cover and glacial erosion might make the use of the 1:20 000 scale more successful in other study areas.

The bedrock within the 1:200 000 scale target area lacks precursor fabrics caused by tectonic deformation as the batholith was emplaced after the Svecofennian orogeny (Rämö and Haapala, 2005). Consequently, we do not expect the fracture or lineament pattern to be controlled by local ductile anisotropies, such as foliations and folds. This simplifies the multi-scale analysis of the fractures and lineaments as investigating the controlling effect of such structures is not required. We do not expect the 1:10 scale fracture lengths to be stratabound as the lithology is homogeneous crystalline rock within the entire 1:200
000 scale target area (Figure 1). However, we cannot have the same expectation for lineaments, especially digitised in the scale 1:200 000, as their interpreted lengths can span tens of kilometers (Table 3) and they are therefore comparable in size to the sheet-like bodies of rapakivi granite with estimated thickness of circa 5-10 kilometer (Rämö and Haapala, 2005). The possible partly stratabound nature of lineaments might be noticeable in their length distributions where the lognormal distribution fit to the 1:200 000 scale trace length distribution is better with a $p$ value of less than 0.1 in the comparison of power-law and
lognormal fits indicating high statistical significance of the lognormal preference (Table 5; Figure 3).

Glacial flow on the Åland Islands has a preferred trend of roughly N-S (Figure 1). Therefore, we expect that the digitised lineaments which are oriented roughly N-S are partly affected by the glacial landforms e.g., in the form of enhancing their length as discussed in a similar study by Ovaskainen (2020). The N-S oriented lineaments are, based on visual observation of Figures 3A and 4, quite continuous and the effect of glacial erosion is apparent from the LiDAR raster maps in the form
of visible linear quaternary land features such as possible roches moutonnées. Furthermore, the N-S striking lineaments are determined to form a distinct azimuth set (Table 4). Inspection of the length distributions of the N-S oriented lineaments using power-law modelling (Table 5) shows that the N-S striking lineaments have the lowest exponents compared to other sets or to all lineaments within both the 1:20 000 and 1:200 000 scales with values of -2.33 and -1.22, respectively. However, the difference is small compared to other sets for the 1:200 000 scale and the sample counts of lengths within each set is low enough to
possibly affect the reliability of the results. Overall, the glacial flow is more than likely a controlling factor in the N-S striking lineament characteristics, but the impact of this factor cannot be quantified. Some evidence of the bedrock-related nature of N-S trending lineaments is present as local fracture azimuth maxima in individual Getaberget target areas and in surveyed fault data (Figures 6 and 9 by Ovaskainen et al., 2022). Detailed geophysical studies can verify the existence of bedrock structures represented by lineaments but their type characterisation is only possible with drilling. The lack of a dominant N-S set of
fractures in the aggregated Getaberget trace data (Figure 5A; Table 4) further brings into question whether the N-S striking lineaments truly represent bedrock structures. The N-S trend is not visible in the geophysical magnetic or electromagnetic rasters either (Figure 3BC). In conclusion, through our multi-scale cross-validation, the inclusion of the N-S striking lineament





characteristics in e.g., discrete fracture network -modelling should be done with caution and the verification of the existence
of N-S striking bedrock structures should be conducted in further studies. Some evidence for the bedrock-related nature is
provided in the existence of a dextral faults trending N-S (Figure 9 by Ovaskainen et al., 2022).

## 5.3   Multi-scale Analysis

Within all subfigures of Figure 6 the length distributions seem to follow the trend of the fitted power-law to some degree,
although both the slope and location (below or above the power-law trend) of individual distributions vary. The differences
in slope compared to the fitted multi-scale power-law can be explained for the 1:10 and 1:20 000 scales by the individual
power-law exponents of circa -2.0 that deviate clearly from the 1:200 000 scale exponents of circa -1.2. The difference in
location could indicate problems with the normalisation of the trace length distributions in the multi-scale plot. On closer
inspection of the 1:10 length distributions in Figure 6, the trend that has been fitted to the lineament trace lengths could fit
the center part (which has lengths below the cut-off) of the 1:10 length distributions. If the cut-off was around 1 $m$ that part
would be included in the fitting and the trend would have a better continuation, at least visually. However, the head (highest
length traces) of the 1:10 distribution would still not fit the multi-scale power-law, possibly indicating the need of both a tail
and head cut-off. The heads of both 1:20 000 and 1:200 000 scale distributions similarly deviate from the trend. The use of an
optimisation algorithm that considers all distributions and chooses cut-offs, possibly for both head and tail, to fit a single multi-
scale power-law trend rather than determining only the tail end cut-off from the individual length distributions could majorly
improve the process while still allowing the full reproducibility of the fitting process. However, the option also remains that the
fractures and lineaments have scaling properties that correlate with the scale of observation rather than having common ones
(Kruhl, 2013; Davy et al., 2010). The possibility of using normalisation methods other than area-normalisation should also be
simultaneously investigated (Bonnet et al., 2001) and the use of the probability density function in place of the complementary
cumulative number might have more merit when analysing multi-scale length data (Bour et al., 2002). The occurrence of
partly scale-independent azimuth sets (Figure 5; Table 4) within our data might be indicators of hierarchical organisation of the
fracture network where the different sets cause differences in the scaling laws between scales of observation, similar to a study
by Ceccato et al. (2022) where this option was discussed for their multi-scale fracture and lineament dataset with scale-variant
azimuth sets.

The topological characteristics of the multiple scales follow a set trend where the *Connections per Branch* values decrease
when the scale increases from 1:200 000 to 1:10 (Figure 7; Table 3). A very similar trend was observed in a multi-scale study
done in the Loviisa region, south-east Finland, within a crystalline rapakivi batholith (Ovaskainen, 2020). The trend could be
the result of e.g., source raster differences (Nixon et al., 2012) or possible differences in the actual topological characteristic
differences between fractures and lineaments of different scales. Another option related to the raster differences is the possible
difficulty or subjective bias in identifying two Y-nodes in cases where they are close to each other and instead labeling the
intersection as a single X-node (Andrews et al., 2019). In any case, the possibility of this kind of trend should be kept in mind
when determining topological characteristics from only a single scale of observation as the value might only represent features
within that observation scale. Based on the high proportion of X-nodes for the 1:200 000 scale lineaments the connectivity





of the fracture network would be estimated to be higher than the estimated connectivity from the 1:10 scale fractures, at least based on this strictly two-dimensional analysis. Future studies could include the estimation of the sub-horizontal fracturing, detected in field surveys (Skyttä et al., 2022), on the connectivity to extend the analysis to the third dimension.

## 6  Conclusions


– Based on azimuth analysis of our study area, that covers most of the Åland Islands mainland, the regional fracture pattern is dominated by WNW-ESE and N-S oriented lineaments. In particular, the WNW-ESE oriented lineaments can be expected to correspond to large brittle bedrock structures that cut the Åland rapakivi batholith as they are prominent in geophysical rasters.


– Using the scale of observation of 1:20 000 we generated lineament data within the 10 to 500 $m$ interval from which brittle structure data is lacking in past studies. However, although we produced lineament data within this range, we found that the lineament lengths did not fit the same power-law trend as lineaments with higher lengths, even within the same scale, indicated by the power-law cut-off of circa 3500 $m$. Further investigation of methods to address this length gap is therefore still required.


– The length distribution analysis of traces of each scale, results in power-law exponents of -2.09, -2.26 and -1.14 for the 1:10, 1:20 000 and 1:200 000 scales, respectively. However, lognormal trends are statistically more likely for all three scales, which causes high uncertainty in whether the power-law exponent results are significant. A common power-law exponent fitted to all scale length distributions simultaneously has an exponent of -1.13. However, using only tail cut-offs, the 1:10 scale fractures and 1:20 000 scale lineaments do not adequately fit the common power-law trend while the


1:200 000 scale lineaments fit it better.

– A trend is observed where the 1:10 scale outcrop fractures show a lower degree of X-nodes and values of *Connections per Branch* compared to lineaments from scales 1:20 000 and 1:200 000. Furthermore, the 1:200 000 scale lineaments have the highest degree of X-nodes and values of *Connections per Branch*. This kind of trend in a theoretically scale-independent characteristic should be kept in mind in future studies, especially when restricted to a single scale of


observation.

– The methodological development related to multi-scale fracture network characterisation displayed in this paper is freely available as part of the open-source FRACTOPO package. As a recommendation for future method development: The methodology around multi-scale length distributions requires further development and e.g., the development of an algorithm for the purpose of automatic cut-off optimisation. We welcome all contributions and discussion related to our open


and freely available code and methods on GitHub.



*Code and data availability.* Source code for the main fracture network analysis software used, *fractopo* v0.5.1, is available free and openly on GitHub (https://github.com/nialov/fractopo/tree/v0.5.1) and Zenodo (Ovaskainen, 2022b) and is licensed with the permissive MIT license. Most used data, including Getaberget shoreline fracture trace data, and the code for specific analyses and creation of figures related to this paper is available on Github (https://github.com/nialov/multi-scale-fracture-networks-aland-islands-2022; master branch) and Zenodo
(Ovaskainen, 2022a). The geophysical rasters are not released publicly due to being commercial datasets of the GTK. As previously released as part of another study Ovaskainen et al. (2022), the orthomosaics from the Getaberget shoreline are available on Zenodo (https://doi.org/10.5281/zenodo.4719627).

## Appendix A:  Appendix - Other magnetic rasters

The total field DGRF-65 grayscale and tilt derivative DGRF-65 grayscale magnetic maps are displayed in Figure A1.

*Author contributions.* Nikolas Ovaskainen: Conceptualisation, Methodology, Software, Investigation, Writing - Original Draft. Pietari Skyttä: Investigation, Writing - Original Draft. Nicklas Nordbäck: Conceptualisation, Methodology, Investigation, Writing - Original Draft. Jon Engström: Investigation, Writing - Original Draft.

*Competing interests.* The authors declare that they have no known competing financial interests or personal relationships that could have appeared to influence the work reported in this paper.

*Acknowledgements.* We acknowledge the Finnish Research Programme on Spent Fuel Management (2019–2022) and Geological Survey of Finland for funding the KYT KARIKKO -project, Mira Markovaara-Koivisto for geophysical lineament interpretation and integration and Sabina Kraatz for partly interpreting the 1:200 000 scale LiDAR lineaments. We further acknowledge the constructive reviews by XXX and XXX and the editorial handling by XXX.



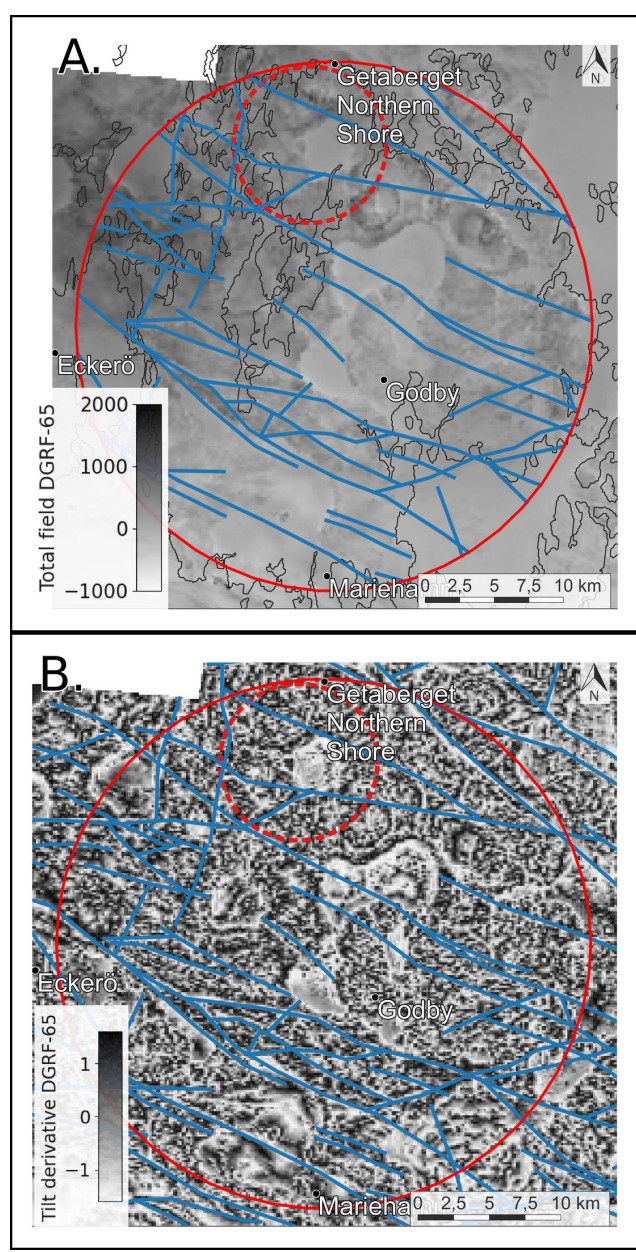

**Figure A1.** Magnetic lineaments that were interpreted using all three magnetic maps (sharp-filtered DGRF-65, Total field DGRF-65 and Tilt derivative DGRF-65) overlay both of the subfigures. A. Total field DGRF-65 grayscale magnetic map used in magnetic lineament interpretation. B. Tilt derivative DGRF-65 grayscale magnetic map used in magnetic lineament interpretation.





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
