# Peer review of "Detailed investigation of multi-scale fracture networks in glacially abraded crystalline bedrock at Åland Islands, Finland"

_EGUsphere, 2022_

## Author Response (AR1)

```
Dear Reviewer 1,

Thank you for the constructive review and comments. Specifically, the
criticism on the lack of clear motivation in the introduction resulted in
a major revision of the introduction to better highlight the motivation and
to link the contents of the introduction to the rest of the paper. See
below for specific responses to your comments. All comments are numbered to
allow referencing from other responses.
Line number references are to the PDF document with the changes highlighted.

On behalf of all the authors,
Turku, Finland, April 2023, Nikolas Ovaskainen
```

The paper is well written, the methods up-to-date and the related results very
interesting and statistically robust. The methods are adequately described, the
limitation discussed and their reliability and representativeness of the obtained
results considered as well. However, some further discussions on methods/results
reliability and robustness are needed.

There is a main problem which undermines the possibility to publish the present
manuscript in Solid Earth: the aim of the research is not clearly stated and thus
it is not clear if the presented methods and results are significant or have any
relevant implications for the case study or what are the broader implications
for the general analyses of lineament maps from remote sensing. Stating clearly
what are the aims and how the presented methods/results can answer to the
research questions will also help in reorganizing logically the discussion sections.
The paper would be also good if it was clearly focused on the presentation of
the new analytical tools and the results adopted for the characterization of a
case study, but still the case study need to have a clear aim. I think that the
Authors can re-arrange the present manuscript with little effort in order to solve
this main issue, otherwise I would suggest to redirect the manuscript to a more
specialized journal.

**Main issues**

The aim of the research and the conclusions/implications of the results need to
be explained a little bit more, they are not immediately understandable from
reading the present manuscript. Especially in the Abstract, Introduction (where
the aims are sparsely presented in bit and pieces) and in the conclusions. There
is a discrepancy between the "results and conclusions" (sparsely) reported in the
introduction section and the final Conclusions of the paper in Section 6. Are the
methods and results presented here resolutive for the problem they aim to solve?

```
1. We have clarified our motivation and aim primarily in the introduction
and added discussion on data, method and statistical reliability based on
comments by both Reviewers. The methods for multi-scale analysis of
fractures and lineaments have been developed for decades but there still
```

> lacks any easily usable common convention for the analysis. Therefore,
> some "method development" text must be included in the manuscript to
> explain the methods we used. Furthermore, there does not exist any
> multi-scale datasets from Finland outside of our very limited prior
> research. So there is demand for both datasets and method development and
> therefore the focus is on both goals in this manuscript. Multi-scale data
> collection, geologic setting and the analysis all cause uncertainties in
> the results and in the discussion section we therefore have to consider
> all with limited ability to completely separate the discussions about
> them as their effect in the results cannot be separated either.

The cut-off issue needs to be properly discussed in depth. What is the statistical significance of a results based on the analysis of only the 3% of the total dataset? Is it representative of the whole distribution? This needs to be discussed from the methodological point of view (is the analytical method adequate to analyse our dataset?). Is the results based on the 3% of the data representing the entire length distribution?

> 2. This issue was also pointed out by Reviewer 2. See the answer we gave
> there in response number 11.

**Minor comments**

Some part need to be rephrased to be a little bit more concise (the part of the discussion regarding glacial erosion)

> 3. We have made the discussion section more concise around the
> discussion on the effect of glacial erosion.

Reconsider the use of "e.g.",

> 4. The copernicus template for latex submissions that we used
> had "e.g." with the comma in numerous examples. If this paper enters the
> typesetting phase we will fix these details as instructed.

Rephrase the section Lines 92-99 : to be moved to the method section and integrated with the exiting description of the data source.

> 5. We moved much of the repeated information about the data to the
> method section where we added any missing information that was
> only in the introduction. See lines 120-136 in the introduction.

Line 206: "?"?

> 6. The reference is fixed to Rohrbaugh et al. 2002 as was intended.

Additional fixes unrelated to specific referee comments:

- Modified Figure 3 to include the second, missing, eastern 1:20 000 circular target area.
- Fixed small reference errors.

- Modified all figure sub-labels to use "(a)", "(b)", ... instead of "A.", ... in figures captions and in text.
- Increased font size in Figure 4.
- Headers were fixed to only have the first letter capitalized.

Dear Reviewer 2, Marco Mercuri,

Thank you for the constructive review and comments. The comments on the data analysis resulted in the additional analysis in the Appendix to satisfy a missing part in our length distribution analysis. Furthermore, it resulted in more clear explanations in the text for our decisions regarding the analysis. Your indepth line-wise comments further resulted in a number of small fixes that overall make the paper more cohesive. See below for more specific responses to your comments. All comments are numbered to allow referencing from other responses.
Line number references are to the PDF document with the changes highlighted.

On behalf of all the authors,
Turku, Finland, April 2023, Nikolas Ovaskainen

**General comments**

The manuscript by Ovaskainen and coauthors deals with a multi-scale analysis of the fracture network affecting the crystalline bedrock at the Åland Islands in Finland. The main aim of the study is to contribute to filling the gap of data sets dealing with multi-scale fracture network analyses of crystalline rocks, which might have important applications, including nuclear waste disposal. The multi-scale approach also allows filling the knowledge gap on the local fracture network at an intermediate scale length range (i.e., 100-500 m). The authors highlight that certain fracture network properties, such as length distribution and connectivity might be scale-dependent. I think this study represents an interesting example of a multi-scale analysis of a fracture network. The study is very well performed, particularly for the collection of a very robust dataset and for the usage of up-to-date methods for fracture analysis, including a new software. Consequently my opinion is that the study deserves to be published in Solid Earth.

A point which does not fully convince me is the analysis of length distribution. As described in lines 249-251 (see also Figure 5) the power law, lognormal, and exponential fit hypotheses for fitting length distribution have all been tested on the range of the lengths (traces or branches) above the power-law cut-off length. For testing the power law fit, I have no doubts on the procedure, but I would like to see also the goodness of fit for exponential and lognormal distributions calculated for the whole data set. Moreover, the data (black dots) below the cut-off are not showed on the figures (Figure 5). Such a procedure also impacts the multi-scale analysis of length distribution (Figure 6). In my opinion, all the cumulative length data should be shown in the Figures 5 and 6. To summarize, I

think that the procedure for the best fitting equation should be revised or better motivated, and some Figures (5 and 6) should be revised accordingly.

> 1. We acknowledge the issue of representation for all length data in the individual scale length distribution plots (Figure 5). The truncated data, i.e. lengths below the cut-off, are not well represented. This is intentional and it is done to focus the plots on the data above the cut-off where all the statistical fits are done. We remedy this lack of representation in this revision by adding a figure in the Appendix with all length data and associated lognormal and exponential fits. However, such analysis is difficult to tie to multi-scale analysis without confusion for the readers (and writers) and the fits are not comparable to the fits done to cut-off truncated data (Clauset et al. 2009; lines 346-348) and
> consequently, the discussion around the appendix results is minimal.
>
> In the multi-scale plot (Figure 6), all length data is shown. The length data points above the cut-offs are colored while the greyed out points represent the data below the cut-offs. Some data points are covered by others.

I suggest publication in Solid Earth after moderate revisions. Please find below specific comments and technical corrections.

Marco Mercuri

**Specific comments**

The Introduction section is well written and the contents are appropriate for this work. However, I find it quite long and composed by a single sub-section (Review). I suggest reorganizing the Introduction section into subsections and, if possible, reducing its length to better highlight the main aim of the work. Perhaps some parts could be moved to the Discussion section.

> 2. We have revised the introduction based on this comment and Reviewer 1 comments. Text was made more concise and some was moved to the method section to remove repetition. The length has been reduced and the research questions and aims are better highlighted in the new structure. The introduction starts with a review subsection which is followed with the "Agenda of our study" subsection which should make apparent the aims of this work. The word count of the introduction was shortened from circa 1400 words to circa 1100 words.

Very interesting part of the introduction at lines 56-64

Line 83; 103-104; Please refer to literature and/or briefly explain with text or a table or a figure the different topological parameters (connection per branch, fracture intensity, dimensionless intensity ...)

```
3. We added references to the parts in the introduction where we refer
to these parameters.
```

I am not an expert in this area, so I find lines 127-130 not easily understandable.
What is the kinematics of the E-W faults? I assume it is left-lateral from the
presence of NE-SW extension fractures in their damage zone. Is it?

```
4. We clarified the text to specify the varying kinematics of the E-W
faults, and specified that the sinistral faults were associated with
the extension fractures, as you pointed out.
```

Data & Methods. Looking at the different scales of observation, I noticed that
a scale similar to 1:1000 has not been considered. I suppose a "basemap" for
mapping fractures at such a scale does not exist for the study area, and perhaps
this should be stated in the text. Just a curiosity: have you considered the usage
of Bing Maps/Google Maps, or is there too much vegetation?

```
5. We have tried using satellite images from different sources and found
Google Maps to be the most accurate and easily available dataset.
However, as you pointed out, the vegetation is the problem. We are only
able to use the images in almost perfectly exposed areas without any
vegetation. In comparison, in LiDAR data the vegetation is (mostly)
filtered so it can be used in more extensive areas.
We have clarified this in text on lines 441-451.
```

Table 1. What do you mean by "cell size"? Please explain it better in the text

```
6. We clarified "cell size" on line 189.
```

Table 3. What is the difference between Number of Traces and "Number of
Traces (Real)". Please, explain it in the caption or in the text.

```
7. We added the "b" symbol to the parameter and explained it in the caption
of Table 3.
```

Table 3. I suggest accompanying the absolute number of each type of node and
branch with the relative percentage for higher clarity.

```
8. We have included the relative percentages of node and branch counts
in Table 3.
```

L295. "Same trend is seen with Dimensionless Intensity B22". The values of B22
are quite similar at different scales, but the trend of P21 with scale is actually
the opposite. Please fix the sentence.

```
9. This was an error and we fixed the sentence.
```

L295-297. "Connections per Trace and Connections per Branch display a trend
with values decreasing as the scale increases with the 1:10 scale having the lowest
value". I agree, but values of connection per Branch at 1:20.000 and 1:200.000
scales are actually pretty similar (1.75 and 1.85). I suggest underlying this point.

> 10. We have noted this point in text but did also point out that the range of
> values for Connections per Branch is limited between 0.0 and 2.0. This
> amplifies these numerically small differences so the difference should be
> considered as at least somewhat significant.

L316-317 and Fig. 5. "The cut-off proportion (i.e., amount of data removed by
the application of the cut-off) for the 1:10 scale is very high with 97.82 % of
data being cut off." Maybe I do not understand this sentence correctly. Do you
mean the power law fit is performed on the remaining data (i.e. about 2% of
data), or that the fit is performed on 97.82% of data? I am not sure that a fit
performed on 2% of data is statistically significant.

> 11. The fits are performed on only ~2 % of the fracture data so your
> interpretation is correct. The majority of
> the digitized fractures are seemingly affected by the
> resolution truncation effect (Pickering et al. 1995). Though this makes
> the fits statistically very uncertain we wished to be consistent in the
> analysis we performed for all data. Even though the fracture length
> distribution by itself might not adequately follow a power-law, it
> does not mean that in a multi-scale length analysis they could not still
> follow a common trend with the lineament data, at least partly. Therefore
> we wished to keep the analysis results for the individual scales to use
> as reference to the multi-scale results. Furthermore, another motivation
> for the use of the power-law comes from the physical rationale (Bonnet et
> al. 2001) that fractures follow some scale-independent and possibly
> fractal law as we mention in text on lines 319-321.

Have you tested the exponential and lognormal fit on the whole dataset, or only
on lengths longer than the power law cut-off length? If not, my suggestion is to
compare the PL fit as it is with the exponential and lognormal fit performed on
the whole datasets.

> 12. We refer to Clauset et al. 2009 on why we cannot compare the fits to
> the full data to the fits on the truncated data. Clauset et al. 2009
> state that it is not statistically valid to compare fits that have been
> conducted on different data. However, we added both a figure and a table
> in the appendix which contains the lognormal and exponential fits to the
> full length data for both traces and branches (See also response 1.).

L363-365: "This cut-off can be estimated to be the lowest length lineaments which
we can consistently interpret without truncation effects caused by resolution of
the LiDAR DEM, assuming that the lineament trace lengths follow a power-law."
In my opinion, such an assumption should be better justified.

> 13. We have expanded the discussion around this assumption after the sentence
> which you pointed out on lines 428-433.

Section 5.3. I think that all the data should be shown on the plots and not only
the data which are above the cut-off power law length. I think that (and it is

also somewhat stated in section 5.3), as presented now, the effect of the left tail (truncation) on the potential multi-scale fit is considered (too much?), but there is also a censoring effect due to the fact that the cumulative number of longer fractures is affected by the size of the sampling window. My suggestion is to plot all the data, trying to manually fit them with a power-law which is tangent to the central part of each distribution (see also Ceccato et al., 2022).

> 14. See response 1. in regards to the missing data
> points (all data should be visible in the multi-scale plot). You are
> correct that the censoring effect of the sampling area also affects the
> results. We touch this issue on lines 522-527 in the "Multi-scale
> analysis" discussion chapter and issue recommendations for method
> development. We are skeptical on the use of a manual method to fit the
> power-law to the multi-scale data as these kinds of methods lack
> reproducibility and are not statistically consistent. Instead, as we
> discuss on lines 527-530, we recommend the development of a
> multi-scale fitting method which considers both the head and tail of the
> distribution. Furthermore, we recommend the use of the density
> distribution in place of the complementary cumulative number that we use
> in this paper. We do however still claim that there is some use in
> publishing the results that we have as is, as the complementary
> cumulative number method is used in many other studies and the results
> are therefore comparable to them and are more reproducible without
> manual fitting.

L467-469. In my opinion, this option is very reliable. Isn't it part of the "source raster differences" (L466)?

> 15. We did not fully understand your point here, as we do already state
> "Another option related to the raster differences ..." i.e. we already
> link the option to the source raster differences. We do not wish to
> amplify this option in our study as we provide no examination of its
> possibility in our digitizations but just point it out from prior
> studies.

Conclusion no 4 (Lines 491-495). OK, but you also said before that such type of analysis can be affected by a lot of biases. In my opinion, it's hard to imagine that the connectivity of a fracture network decreases with increasing detail of the analysis. I suggest removing or toning down this conclusion.

> 16. We appended the conclusion with the suggestion that it might be related
> to methods rather than nature itself. This is the point we wished
> to bring out and it should be more clear now thanks to your suggestion.

**Technical corrections**

Line 28. Consider using "multiple" instead of "multi-scale" to improve readability.

> 17. We changed the word to "multiple" as per your suggestion and

```
brought up the importance of using multiple scales.
```

Lines 35-36: There is repetition from Line 27. Please remove one of them.

```
18. We removed the repeated lines which were originally at 35-36.
```

Line 111: The wording "The main part of Åland Islands bedrock is comprised of the 1.58 Ga Åland Batholith" is odd. Please rephrase.

```
19. We rephrased to clarify that we mean by the main island.
```

Figure 1: The text in the image is very small. Perhaps enlarging the figure in the final version could fix this.

```
20. We enlarged the figure in the pdf and the text is consequently more
clear.
```

Line 132 (and in many other parts): "e.g." should not be capitalized.

```
21. Thank you, we fixed the capitalizations.
```

Line 138: Consider using "As an example" or similar instead of "e.g." at the beginning of the sentence.

```
22. We have changed some uses of "e.g.", particularly at beginnings
of sentences to use "for example". This should improve readability.
```

Lines 141-142: Please use consistent nomenclature here with Table 1 for better clarity.

```
23. We have added clarifications both to the Table 1 caption and to the text.
We specify by what terms we refer to the resolution and areal extent.
However, we believe including the specific terms we use ("Cell Size" and
"Total Target Area") along with the more general terms is justified.
Resolution is a more general term compared to the "Cell Size" which more
specifically states how we defined the resolution for raster datasets.
We also refer to "target areas" many times in text.
```

Line 206: There is a "?" in the brackets. Please remove it.

```
24. We fixed the citation to Rohrbaugh et al. 2002 as was originally
intended.
```

Lines 215-217: Please use P22 instead of P21.

```
25. We fixed the reference from "Dimensionless Intensity P21" to "Dimensionless
Intensity P22" as you pointed out!
```

Figure 3: The white text in the image is not easily readable. Please change the color of the text.

```
26. We changed to color of the labels in Figure 3 to have a orange foreground.
This makes them stand out more in the images. We also increased
the size of the points that were associated with the labels as they
were almost imperceptible before as you pointed out.
```

Figure 6: Please provide an explanation of the acronyms (e.g., ANCCM) in the caption.

    27. We included explanations of both ANCCM and MSLE in the Figure 6 caption.

Line 392: Please rephrase as "[. . . ] with high geophysical but low topographic signals."

    28. We removed the "and" word to fix the sentence.

Lines 481-483: Please rephrase the sentence to be more concise.

    29. We made the conclusion more concise as suggested.

References: Please pay attention to the formatting of some references (e.g. capital letters for the reference at Line 610).

    30. We fixed the capitalized letters in the titles of some references including the referred Middleton et al. 2015 reference on line 610.

Additional fixes unrelated to specific referee comments:

- Modified Figure 3 to include the second, missing, eastern 1:20 000 circular target area.
- Fixed small reference errors.
- Modified all figure sub-labels to use "(a)", "(b)", . . . instead of "A.", . . . in figures captions and in text.
- Increased font size in Figure 4.
- Headers were fixed to only have the first letter capitalized.

---

## Referee Report (RR1)

**Revision of "Detailed investigation of multi-scale fracture networks in glacially abraded crystalline bedrock at Åland Islands, Finland" by N. Ovaskainen and coauthors**

**General comments**

Dear Editor and Authors,

The manuscript by Ovaskainen and cohautors has been improved from its initial version. In particular, the introduction section is now more concise and organised, consisting in two sub-sections, one of the which focuses on the aims and scope of the manuscript. The cumulative distributions of all fracture lengths for all datasets are now shown in a supplementary figure. I still believe that this is an interesting study for the implications, impressive dataset, and the use of a new software for fracture network analysis. Therefore, in my opinion, the study deserves to be published on Solid Earth. However, I think that the manuscript still requires some revisions in the methodology.

I apologize, but I still have concerns regarding the methodology used for fitting single-scale and multi-scale cumulative length distributions. In my opinion, the reasoning for choosing the cut-off length and the assumption that the cumulative length distributions (above the cut-off length) are best fitted by a power-law is circular. The authors assume a priori that part of the cumulative length distribution can be fitted with a power-law, and as a result, the algorithm returns the power-law parameters and the cut-off length. The authors then only consider the lengths above the cut-off when comparing the power law fit with lognormal and exponential fits, concluding that the power-law fit is the best. Essentially, the cut-off length found in this way is taken as representative of the censoring bias.

I agree that the fit with different equations should be tested on the same range of lengths and that lengths affected by the censoring bias should not be considered in the cumulative length distribution fit. However, I do not understand why the authors only consider that specific range of lengths. Additionally, the fit is performed on only around 2% of the dataset, which may not be statistically representative. Another concern I have is why the authors give more weight to the truncation bias and not the censoring bias.

I suggest two possible solutions to address this issue:

1)  Specify and convince readers that the assumption that cumulative lengths are distributed following a power law and that only lengths above the cut-off are important while the remaining lengths are not affected by a censoring bias.

2)  Revise the methodological part of the manuscript. Here, I see two possible solutions:

    A.  Conduct a comparison between the cumulative length distribution on **all** data (substituting Fig. 5 with the new fig B1 plus the power law fit). The authors can discuss the fact that the power-law can only be applied to a small range of lengths, or alternatively, the cut-off can be set as equal to the minimum trace/branch length. Negative exponential and lognormal distributions fit the cumulative length distribution better (looking at Tab B1 and Fig. B1). Additionally, the fit for the multi-scale cumulative length distribution should be performed

on all data, not just on the data above the cut-off length (i.e. the gray data points in Figure 6).

  B. Apply the Maximum Likelihood Estimators - Kolmogorov-Smirnoff (MLE-KS) test considering various ranges of lengths, as described in some works (e.g., Dichiarante et al., 2020; Ceccato et al., 2022).

I acknowledge that option 2B, although very accurate, would be very time consuming and the topic is not the focus of the work. My suggestion to the authors is to evaluate option 2A, highlighting that the effect of censoring and truncation has been not considered in the fitting procedure.

The number of lines refers to the clean version of the manuscript.

Marco Mercuri

**Specific comments**

1. Lines 89-92: Yes, but please anticipate here that the decrease in connectivity with increasing scale could be a a methodological issue.
2. Lines 363-366 These lines should be revised depending on whether and how the authors decide to revise the methodology.
3. Line 374. The authors might be interested in a recent paper which shows the use of Bing Maps for fracture network characterisation: Mercuri, M., Tavani, S., Aldega, L., Trippetta, F., Bigi, S., and Carminati, E. (2023). Are open-source aerial images useful for fracture network characterisation? Insights from a multi-scale approach in the Zagros Mts. Journal of Structural Geology, 104866
4. Lines 489-493 These lines should be revised depending on whether and how the authors decide to revise the methodology.

**Technical corrections**

- Lines 9-10: "The best fit to model the lineaments and fracture lengths with a common power-law resulted in an exponent of -1.13". The sentence is not easily understandable during a first read. Please rephrase it.
- Lines 11-12: I suggest removing "could"
- L 24-25 There is a repetition of lines 19-20. I suggest removing these lines.
- I suggest rephrase this sentence into something like "requires collecting fracture and lineament data using a combination of methods and preferably from multiple scales of observation" for improving conciseness.
- L76 Please remove "e.g."
- L126. I suggest changing "e.g." with "due, for example, to" or something similar.
- L144. Maybe the reference in the text should be written like "published by Ovaskainen et al. (2022)". Please check the guidelines for authors.
- L217. Please rephrase in "represent non connected nodes" for better clarity

---

## Author Response (AR2)

Dear Editor, Stefano Tavani, and Reviewer, Marco Mercuri,

Thank you again for the constructive review. The criticism related to our lack of focus on censoring is valid. Previously the effect of censoring was mostly mentioned only on lines 494-499 in the discussion. Consequently, we have included some new minor analysis results related to the effect of censoring on our trace datasets. The new results provide alternative values for e.g. power-law exponents and allow critical assessment of the previous results. However, they do not override the previous results which are thus still kept intact.

See below for specific responses to your comments. Line number references are to the PDF document with the changes highlighted.

Other changes which might not be highlighted in the PDF:

- Reference to Skyttä et al. 2023 has been promoted from a preprint to the published article
- Appendix Table B1 has been updated with power-law exponent values.
- "Fig." is used in place of "Figure" and "Figs." in place of "Figures" as instructed in the author guidelines
- Addition of Fig. 6 with censoring investigation

On behalf of all the authors, Turku, Finland, May 2023, Nikolas Ovaskainen

Revision of "Detailed investigation of multi-scale fracture networks in glacially abraded crystalline bedrock at Åland Islands, Finland" by N. Ovaskainen and coauthors

General comments

Dear Editor and Authors,

The manuscript by Ovaskainen and cohautors has been improved from its initial version. In particular, the introduction section is now more concise and organised, consisting in two sub- sections, one of the which focuses on the aims and scope of the manuscript. The cumulative distributions of all fracture lengths for all datasets are now shown in a supplementary figure. I still believe that this is an interesting study for the implications, impressive dataset, and the use of a new software for fracture network analysis. Therefore, in my opinion, the study deserves to be published on Solid Earth. However, I think that the manuscript still requires some revisions in the methodology.

I apologize, but I still have concerns regarding the methodology used for fitting single-scale and multi-scale cumulative length distributions. In my opinion, the reasoning for choosing the cut-off length and the assumption that the cumulative length distributions (above the cut-off length) are best fitted by a power-law is circular. The authors assume a priori that part of the cumulative length distribution can be fitted with a power-law, and as a result,

the algorithm returns the power- law parameters and the cut-off length. The
authors then only consider the lengths above the cut- off when comparing the
power law fit with lognormal and exponential fits, concluding that the
power-law fit is the best.

We now highlight the inability to fit a power-law to the full data in the appendix
Fig. B1, reducing the "circular logic" mentioned. We would like to note that
we do not conclude that the power-law fit is best for any of the scale length
distributions. Rather, the lognormal fit is the best fit for all length data,
regardless of scale or categorization into azimuth sets (see R-values in Table 5).
However, the significance of the goodness-of-fit determination (p-values) vary,
providing avenues of comparisons between the data. Furthermore, the multi-
scale nature (i.e. power-law/fractal) of bedrock fractures is well documented and
is based on physical rationale. We therefore believe providing the power-law
characteristics, even if power-law is not the best statistical fit, has value.

Essentially, the cut-off length found in this way is taken as representative of
the censoring bias. I agree that the fit with different equations should be
tested on the same range of lengths and that lengths affected by the censoring
bias should not be considered in the cumulative length distribution fit.
However, I do not understand why the authors only consider that specific range
of lengths. Additionally, the fit is performed on only around 2% of the
dataset, which may not be statistically representative. Another concern I have
is why the authors give more weight to the truncation bias and not the
censoring bias.

I suggest two possible solutions to address this issue:

1) Specify and convince readers that the assumption that cumulative
   lengths are distributed following a power law and that only lengths
   above the cut-off are important while the remaining lengths are not
   affected by a censoring bias.

2) Revise the methodological part of the manuscript. Here, I see two
   possible solutions:

   A. Conduct a comparison between the cumulative length distribution
   on all data (substituting Fig. 5 with the new fig B1 plus the power
   law fit). The authors can discuss the fact that the power-law can
   only be applied to a small range of lengths, or alternatively, the
   cut-off can be set as equal to the minimum trace/branch length.
   Negative exponential and lognormal distributions fit the cumulative
   length distribution better (looking at Tab B1 and Fig. B1).
   Additionally, the fit for the multi-scale cumulative length
   distribution should be performed on all data, not just on the data
   above the cut-off length (i.e. the gray data points in Figure 6).

B. Apply the Maximum Likelihood Estimators – Kolmogorov–Smirnoff (MLE-KS) test considering various ranges of lengths, as described in some works (e.g., Dichiarante et al., 2020; Ceccato et al., 2022).

I acknowledge that option 2B, although very accurate, would be very time consuming and the topic is not the focus of the work. My suggestion to the authors is to evaluate option 2A, highlighting that the effect of censoring and truncation has been not considered in the fitting procedure.

We now provide analysis of the effect of a censoring cut-off on the determination of the power-law exponent, cut-off and subsequent cut-off proportion in Fig. 6. This allows critical assessment of the previously determined exponents and the truncation i.e. tail cut-offs. Along with the figure, text has been added to the methods (line 256 onwards), results (lines 351-362), discussion (line 399 onwards) and conclusion (lines 538-545) sections. This approach is a combination of the suggested options 1 and 2A as we both better clarify our reasoning on focusing on truncation cut-offs for the multi-scale length analysis and provide analysis of the effect of censoring in the single scale length distribution analysis. As previously mentioned, this analysis does not rule out the previous analysis but provides e.g. alternative exponent values and avenues of discussion. We have not replaced Fig. 5 with Fig. B1 from the appendix, as suggested in option 2A, as analysis of the full length data is not a focus and we hope the provided text, analysis and reasoning address the pointed out issues.

To summarise our reasoning on the focus on truncation cut-offs here: We firstly lack technical tools to reproducibly optimise for both truncation and censoring cut-offs simultaneously for multi-scale data (lines 277-278 and 502-505). Furthermore, the effect of the tail (low lengths) on the multi-scale fitting process is much more significant than the head due to our use of the cumulative number in the fitting for the multi-scale length data (lines 278-280). To remove the higher significance of the tail lengths in future studies we recommend the use of the probability density function in the discussion section (lines 507-509). Lastly, the defining of a censoring cut-off is less studied in literature as the effect of censoring is less obvious than the horizontal trend related to truncation, which is clearly observed in the tail lengths in log-log length distribution plots.

The number of lines refers to the clean version of the manuscript.

Marco Mercuri

Specific comments

1. Lines 89-92: Yes, but please anticipate here that the decrease in connectivity with increasing scale could be a a methodological issue.

The possible methodological cause is now mentioned on lines 91-92.

2. Lines 363-366 These lines should be revised depending on whether and how the authors decide to revise the methodology.

The lines 399-404 have been revised to include the discussion of censoring.

3. Line 374. The authors might be interested in a recent paper which shows the use of Bing Maps for fracture network characterisation: Mercuri, M., Tavani, S., Aldega, L., Trippetta, F., Bigi, S., and Carminati, E. (2023). Are open-source aerial images useful for fracture network characterisation? Insights from a multi-scale approach in the Zagros Mts. Journal of Structural Geology, 104866

Thank you! We have included it as a reference on line 418.

4. Lines 489-493 These lines should be revised depending on whether and how the authors decide to revise the methodology.

We have included the lacking censoring analysis for the given values on line 538-539 and added a new separate conclusion regarding the censoring investigation on lines 542-545.

Technical corrections

- Lines 9-10: "The best fit to model the lineaments and fracture lengths with a common power- law resulted in an exponent of -1.13". The sentence is not easily understandable during a first read. Please rephrase it.

The sentence has been rephrased to be clearer.

- Lines 11-12: I suggest removing "could"

Replaced with "can".

- L 24-25 There is a repetition of lines 19-20. I suggest removing these lines.

The repeating part ", but it is crucial ... characterisation." was removed.

- I suggest rephrase this sentence into something like "requires collecting fracture and lineament data using a combination of methods and preferably from multiple scales of observation" for improving conciseness.

We rephrased the sentence on lines 33-34.

- L76 Please remove "e.g."

It has been removed.

- L126. I suggest changing "e.g." with "due, for example, to" or something similar.

The sentence has been rephrased to overall be more clear.

- L144. Maybe the reference in the text should be written like "published by Ovaskainen et al. (2022)". Please check the guidelines for authors.

Thanks, the reference was written wrong. It has been fixed.

- L217. Please rephrase in "represent non connected nodes" for better clarity

We rephrased as suggested.